# Lifestyle practices and associated factors among adults with hypertension: Conquering Hypertension in Vietnam-solutions at the grassroots level study

**Phuong H. Nguyen**[1]*, **Robert J. Goldberg**[2], **Jeroan J. Allison**[2], **Diep B. Nguyen**[3,4], **Ha T. Tran**[1], **Oanh M. Tran**[5], **Duc A. Ha**[5], **Hieu L. Nguyen**[6], **Brittany A. Tran**[7], **Bo Wang**[2], **Hoa L. Nguyen**[2]

1 Department of Sociology of Health, Faculty of Public Health, Thai Binh University of Medical and Pharmacy, Thai Binh, Vietnam, 2 Department of Population and Quantitative Health Sciences, University of Massachusetts Chan Medical School, Worcester, Massachusetts, United States of America, 3 Department of Epidemiology, Institute for Preventive Medicine and Public Health, Hanoi Medical University, Hanoi, Vietnam, 4 Center for Training and Research on Substance Abuse-HIV, Hanoi Medical University, Hanoi, Vietnam, 5 Health Strategy and Policy Institute, Hanoi, Vietnam, 6 Hanoi Medical University Hospital, Hanoi, Vietnam, 7 Department of Medicine, University of Massachusetts Chan Medical School, Worcester, Massachusetts, United States of America

* hphuong.950522@gmail.com

**Data Availability Statement:** The authors confirm that, for approved reasons, some access restrictions apply to the data underlying the

## Abstract

### Background

Vietnam is experiencing an increasing prevalence of hypertension in its adult population. In addition to medical therapy, modifying adverse lifestyle practices is important for effective blood pressure control. There are limited data on unhealthy lifestyle practices in patients with chronic diseases, however, particularly among hypertensive patients living in rural Vietnam. Our study objectives were to examine the prevalence of unhealthy lifestyle practices and associated factors among rural Vietnamese adults with uncontrolled hypertension.

### Methods

Data from the baseline survey of a cluster randomized trial among hypertensive Vietnamese adults (2017–2022) were utilized. Information on unhealthy lifestyle practices including smoking, excessive alcohol consumption, physical inactivity, and inadequate fruit and vegetable intake was collected from study participants. The primary study outcome was having ≥2 unhealthy lifestyle practices. A multivariable logistic regression model was used to examine factors associated with the primary study outcome.

### Results

The mean age of the 671 patients was 67 years and 45.0% were men. Nearly three out of every four participants had one or fewer unhealthy practices, 24.0% had two, and 3.3% had three or all four unhealthy lifestyle practices. Men, individuals who did unpaid work or were unemployed, and individuals with hypertension level III were more likely to have ≥2

findings. The data underlying the study are human subject clinical data. Data are available from the Vietnam Health Strategy and Policy Institute (HSPI) Institutional Data Access/Ethics Committee for researchers who meet the criteria for access to confidential data. Requests may be made to Giang Nguyen, Email: nguyengiang@hspi.org.vn.

**Funding:** The authors who were funded included Dr. Jeroan J. Allison, Dr. Duc A. Ha, and Dr. Oanh M. Tran. The present study was supported by the "Training Program for Strengthening Capacity in Non-communicable Diseases- D43TW011394-01" which was funded by the National Institutes of Health/ Fogarty International Center. The authors who were funded included Dr. Jeroan J. Allison and Dr. Hoa L. Nguyen. The funders had no role in our study design, data collection and analysis activities, decision to publish, or preparation of the manuscript.

**Competing interests:** NO authors have competing interests

unhealthy lifestyle practices, whereas individuals with higher education were less likely to have ≥2 unhealthy lifestyle practices compared with respective comparison groups.

## Conclusions

We observed a high prevalence of unhealthy lifestyle practices among rural Vietnamese patients with uncontrolled hypertension. Several demographic factors were associated with a greater number of unhealthy lifestyle practices. Newer interventions and educational programs encouraging lifestyle modification practices are needed to control hypertension among adults living in rural settings of Vietnam.

## Introduction

Vietnam is undergoing an epidemiological transition with changes in disease patterns from infectious to non-communicable diseases, with cardiovascular disease accounting for nearly one-third of all deaths [1]. Hypertension is a primary modifiable risk factor for heart disease and stroke and is an important risk factor for disability-adjusted life-years lost [2, 3]. A national epidemiological survey of adults aged 25 years and older living in Vietnam found that one in every four adults had hypertension during the survey years of 2002–2008. Among person with hypertension, one in every two were aware of their diagnosis, approximately one in every three individuals were receiving treatment, and only one in every nine adults had their hypertension controlled [4]. Results from the Vietnam May Measurement Month campaign in 2019 showed that high blood pressure was present in one out of every three participants [5].

Despite its magnitude, hypertension can be effectively managed with non-pharmacological and pharmacological interventions. Healthy lifestyle practices play an important role in achieving and maintaining blood pressure goals and are often utilized prior to initiation of medical therapy [6, 7]. The management of hypertension through lifestyle modifications is particularly cost-effective for low- and middle-income countries for purposes of preventing the morbidity and mortality associated with this condition. While there have been several studies conducted in Vietnam focusing on the lifestyle practices among patients with hypertension, these studies suffer from several limitations including sampling methods [8] and blood pressure measurements [9]. The present study objectives were to examine the prevalence of, and factors associated with four unhealthy lifestyle practices among adults with uncontrolled hypertension living in rural Hung Yen province, Vietnam.

## Methods

### Study setting

The present study utilized baseline data from the parent trial entitled "Conquering Hypertension in Vietnam-solutions at the grassroots level (2017–2022)", a cluster randomized controlled trial that examined the impact of a multi-component intervention to lower blood pressure among adults living in Hung Yen province with uncontrolled hypertension [10]. The parent trial registration is available on ClinicalTrial.gov NCT03590691. The date when data were accessed for the present study purposes was on January 4, 2022.

In brief, a total of 16 communities from 4 districts in Hung Yen province were selected for this cluster randomized trial. Adults aged 25 years and older living in 8 communities were randomly assigned to the intervention group, while residents living in a similar number of communities were randomized to the comparison group. The intervention and control groups

differed in the management of their elevated blood pressure. Both groups received a multi-level intervention modeled on the Vietnam National Hypertension program. In addition, individuals in communities randomly assigned to the intervention group received three enhancements: expanded community health worker services, home blood pressure self-monitoring, and a storytelling intervention delivered via DVDs [10].

## Study population

To be enrolled in the parent trial, consenting adult men and women 25 years and older needed to satisfy each of the following criteria: be a resident of the selected community; have a diagnosis of uncontrolled hypertension according to the 8th Joint National Commission of High Blood Pressure (JNC 8); [11] not be cognitively impaired; not be a "story teller" used to develop the trial intervention; not be a family member of another participant in the study; and not be pregnant at the time of study enrollment.

## Data collection

The data used for the present study were collected at the time of baseline trial enrollment [10]. Data sources included standardized blood pressure and anthropometric measurements, the World Health Organization (WHO) STEPwise approach to non-communicable disease risk factor surveillance (STEPs) protocol, [12] and medical record review.

Blood pressure and anthropometric measurements were collected by trained community health workers and nurses. Height and weight were measured in the absence of shoes and heavy clothing according to a standardized protocol. Patient's blood pressure levels were measured by using the OmROn HEM-8712 automated blood pressure monitor with special attention to assessment and maintenance of the instrument's accuracy along with training and certification of trial research assistants. Three measurements of blood pressure were taken, separated by at least one minute, and values from the last two measurements were averaged and entered into the study database for purposes of analysis.

An in-person interview was conducted by trained study staff using standardized questionnaires from the WHO STEPs survey to collect information on patient's socio-demographic characteristics and self-reported lifestyle practices.

## Study outcome and other variables

The number of unhealthy lifestyle practices among eligible and consenting adults with hypertension was the primary outcome of the present study. We examined four self-reported lifestyle practices including tobacco use, alcohol consumption, physical activity, and diet. These lifestyle practices were assessed by use of the WHO STEPs survey which asked questions about smoking, alcohol consumption during the past 30 days, and engagement in regular physical activity, with the calculation of metabolic equivalent of tasks (METs) to calculate the total time spent in physical activity, [12] and study participants fruit and vegetable intake.

Smoking was defined as the self-reported use of any tobacco products, such as cigarettes, cigars, or pipes in the past 30 days. Excessive use of alcohol was defined as the consumption of >28 grams of pure alcohol for men and >14 grams of pure alcohol for women on an average day in the past 30 days [13]. Patients who reported participating in less than the equivalent of 600 METs on a weekly basis (150 minutes of moderate-intensity physical activity weekly) were classified as being physically inactive [14, 15]. Inadequate fruit and vegetable intake was classified as eating less than the recommended 5 servings of fruits and vegetables on a daily basis [12, 16]. In the present study, we created a composite study endpoint of the number of unhealthy practices for each study participant (range 0–4).

Socio-demographic, medical history, and clinical factors that were potentially associated with the primary study outcome included age group (25–54, 55–64, 65–74, and ≥75 years), sex, marital status (with and without a partner), education level (primary school or less, secondary school, high school, and college or university), occupation (employed, unpaid work or unemployed, and retired), body mass index (BMI) (underweight: BMI <18.5; normal: BMI between 18.5–22.9; overweight: BMI between 23–24.9; and obese: BMI ≥25.0), [17] and level of hypertension (i.e., level I with systolic blood pressure (SBP) between 140–159 mmHg and/ or diastolic blood pressure (DBP) between 90–99 mmHg, level II with SBP between 160–179 mmHg and/or DBP between 100–109 mmHg, and level III with SBP ≥180 mmHg and/or DBP ≥110 mmHg) [18, 19].

## Data analysis

The principal study outcome of interest was the number of self-reported unhealthy lifestyle practices (range: 0–4). Initially, we classified the number of unhealthy lifestyle practices into four categories: ≤1, 2, 3, and all 4 unhealthy practices. Based of the distribution of data in the study sample, and due to the small sample sizes in several categories (e.g., 0, 3, and 4 unhealthy practices), we analyzed the study as a binary outcome (≤1 and ≥2 unhealthy practices).

Continuous variables were summarized as means (±SD) or medians (IQR) compared between patient groups with ≤1 and ≥2 unhealthy lifestyle practices using t-tests or Wilcoxon rank sum tests. Categorical variables were summarized as frequencies and percentages and compared between patient groups with ≤1 and ≥2 unhealthy lifestyle practices using Chi-square or Fisher's exact tests.

In examining factors associated with having ≥2 unhealthy lifestyle behaviors, any variable that yielded a $p$-value < 0.20 in the univariate analysis was included in a multivariable logistic regression model. The area under the ROC curve (or C-statistics) was calculated for the final multivariable logistic regression model and a Goodness-of-fit test was used to test if the model fit the data. A $p$-value <0.05 was considered as statistically significant.

We also performed a multinominal regression model for the three-category outcome (having 0, 1, and ≥2 unhealthy lifestyle practices). The results from this modeling approach were consistent with those from the binominal logistic regression for the binary outcome (≤1 and ≥2 unhealthy practices). However, due to the relatively small sample size in the "0 unhealthy" group and low frequencies in some response categories, we combined participants with 0 and 1 unhealthy lifestyle. We reported results from the binominal logistic model. All analyses were performed using STATA 17.0 [20].

## Ethical consideration

The Institutional Review Board at the Health Strategy and Policy Institute in Hanoi, Vietnam approved the parent trial (Decision 171/QD-CLCSYT). All patients in this study provided written informed consent.

## Results

### Study population characteristics

The study sample consisted of 671 adult men and women between the ages of 38 and 91 years old with uncontrolled hypertension. The mean age of the study sample was 67.0 years, 45.0% of study participants were men, and 45.8% were employed. Approximately 50.0% had a normal BMI and 46.0% had level I hypertension. At baseline, the mean systolic blood pressure of the study sample was 161 mmHg while the mean diastolic blood pressure was 92 mmHg.

## Prevalence of unhealthy lifestyle practices

Approximately one in every nine study participants stated that they were smoking at the time of survey completion (11.2%) (Table 1). Slightly more than one-quarter (26.1%) of the study population were current alcohol drinkers, and one out of every seven patients who reported drinking alcohol consumed excessive amounts of alcohol. The median MET physical activity score was 28.0 MET-hours/week and nearly one-quarter of study participants were considered to be physically inactive. The median number of fruit and vegetable servings reportedly consumed was 3.0 servings and three-quarters of study participants had inadequate fruit and vegetable intake. Among study participants, approximately 18.0% did not have any unhealthy lifestyle behavior, more than one-half had one unhealthy lifestyle practice (54.8%), approximately one-quarter (24.0%) had two unhealthy practices, and 3.0% and 0.3% had any three or all four unhealthy lifestyle practices, respectively (Table 1).

## Study participant's characteristics according to number of unhealthy lifestyle practices

In examining various socio-demographic and other factors associated with unhealthy lifestyle practices, men and older individuals were significantly more likely to have any two or more unhealthy lifestyle practices than women and younger study participants. Participants with lower education, persons who were unemployed or did unpaid work, and those with hypertension level III were more likely to have two or more unhealthy behaviors relative to respective comparison groups (Table 2).

## Factors associated with having 2 or more unhealthy lifestyle practices

Men were more than two times more likely to have any two or more unhealthy practices as compared with women (OR = 2.25; 95% CI: 1.51–3.35). Individuals who were unemployed or

**Table 1. Lifestyle practices among study participants (n = 671).**

|  | n | % |
|---|---|---|
| Smoking | | |
| Current smokers | 75 | 11.2 |
| Alcohol consumption | | |
| Current drinkers (last 30 days) | 195 | 26.1 |
| Participants with harmful use of alcohol | 25 | 3.7 |
| Physical activity | | |
| Time spent on physical activity on a typical week (median, IQR), MET-hours/week | 28.0 (13.2–67.0) | |
| Participants with physical inactivity | 151 | 22.5 |
| Fruit and vegetable intake | | |
| Number of fruit and vegetable servings on an average day (median, IQR), servings/day | 3.0 (2.0–4.8) | |
| Participants with inadequate fruit and vegetable intake | 507 | 75.6 |
| Number of unhealthy lifestyle practices | | |
| None | 120 | 17.9 |
| One | 368 | 54.8 |
| Two | 161 | 24.0 |
| Three | 20 | 3.0 |
| All four | 2 | 0.3 |

IQR: Inter quantile range.

**Table 2. Study participant characteristics according to number of unhealthy lifestyle practices.**

| | Number of unhealthy lifestyle practices | | *p*-value |
|---|---|---|---|
| | ≤1 | ≥2 | |
| Age (mean±SD) (years) | 66.3±8.7 | 68.1±9.5 | **0.018** |
| Age group (years) (n, %) | | | |
| 25–54 | 41 (8.4) | 9 (4.9) | **0.082** |
| 55–64 | 157 (32.2) | 58 (31.7) | |
| 65–74 | 205 (42.0) | 70 (38.3) | |
| ≥75 | 85 (17.4) | 46 (25.1) | |
| Male (n, %) | 202 (41.4) | 100 (54.6) | **0.002** |
| Marital status (n, %) | | | |
| With a partner | 387 (79.3) | 262 (78.1) | 0.74 |
| Education level (n, %)* | | | |
| Primary school or less | 114 (23.4) | 58 (31.9) | **0.032** |
| Secondary school | 267 (54.8) | 87 (47.8) | |
| High school | 64 (13.2) | 29 (15.9) | |
| College/University | 42 (8.6) | 8 (4.4) | |
| Occupation (n, %) | | | |
| Employed | 242 (49.6) | 65 (35.5) | **<0.001** |
| Unpaid work/Unemployed | 140 (28.7) | 83 (45.4) | |
| Retired | 106 (21.7) | 35 (19.1) | |
| BMI category (n, %) | | | |
| Underweight | 23 (4.7) | 8 (4.4) | 0.48 |
| Normal | 221 (45.3) | 95 (51.9) | |
| Overweight | 137 (28.1) | 43 (23.5) | |
| Obese | 107 (21.9) | 37 (20.2) | |
| HTN level (n, %) | | | |
| Level I | 233 (47.7) | 76 (41.5) | **0.008** |
| Level II | 201 (41.2) | 70 (38.3) | |
| Level III | 54 (11.1) | 37 (20.2) | |

SD: Standard deviation, BMI: Body mass index, HTN: Hypertension

*Missing data in 2 study participants

who did unpaid work were more than two times more likely to have any two or more unhealthy practices as compared with employed individuals (OR = 2.28, 95% CI: 1.45–3.58). Participants whose education level was a secondary school or college/university were 44.0% (OR = 0.56, 95% CI: 0.34–0.91) and 66.0% (OR = 0.34, 95% CI: 0.13–0.85) less likely to have any two or more unhealthy lifestyle practices as compared with those with primary school or less. Study participants with hypertension level III were nearly two times more likely to have any two or more adverse behaviors as compared with those at level I (OR = 1.86, 95% CI: 1.11–3.10) (Table 3).

The goodness-of-fit test of the final logistic regression model yielded a *p*-value of 0.64 and the Area under the ROC curve was 0.67.

## Discussion

Hypertension is a growing public health concern in Vietnam. In addition to medical therapy, lifestyle modifications are important for controlling an individual's elevated blood pressure.

**Table 3. Factors associated with ≥2 unhealthy lifestyle practices.**

| | Adjusted OR (95% CI) | *p*-value |
|---|---|---|
| Age group (years) | | |
| 25–54 | 1.0 | |
| 55–64 | 1.55 (0.69–3.46) | 0.29 |
| 65–74 | 1.05 (0.46–2.41) | 0.91 |
| ≥75 | 1.24 (0.50–3.10) | 0.64 |
| Sex | | |
| Female | 1.0 | |
| Male | **2.25 (1.51–3.35)** | **<0.001** |
| Education level | | |
| Primary school or less | 1.0 | |
| Secondary school | **0.56 (0.34–0.91)** | **0.020** |
| High school | 0.74 (0.39–1.41) | 0.36 |
| College/University | **0.34 (0.13–0.85)** | **0.022** |
| Occupation | | |
| Employed | 1.0 | |
| Unpaid work/Unemployed | **2.28 (1.45–3.58)** | **<0.001** |
| Retired | 1.22 (0.70–2.11) | 0.47 |
| HTN level | | |
| Level I | 1.0 | |
| Level II | 1.07 (0.72–1.57) | 0.74 |
| Level III | **1.86 (1.11–3.10)** | **0.017** |

HTN: Hypertension, OR: Odds ratio, CI: Confidence interval

The results of our study show the relatively high prevalence of unhealthy lifestyle practices among hypertensive patients in rural Hung Yen province, Vietnam. Factors including male sex, lower education level, working without getting paid or being unemployed and having hypertension level III were associated with having any two or more unhealthy lifestyle practices.

## Prevalence of unhealthy lifestyle practices

**Smoking.** More than one in every nine participants in the present study were currently smoking, an estimate which is similar to the frequency of current smokers reported in previous studies in Vietnam [8, 21]. There is a known risk between smoking and higher blood pressure findings [22, 23], further reinforcing the importance of smoking cessation in order to achieve blood pressure goals.

**Alcohol consumption.** Approximately one-quarter of our study population currently consumed alcohol on a regular basis. This finding is consistent with the results from two previous studies in Vietnam [8, 21] which showed a high frequency alcohol consumption among hypertensive patients. Excessive use of alcohol was the unhealthy lifestyle practice of the lowest percentage in the present study (3.7%), which was lower than data reported from 400 adults aged from 40 to 69 years in Thua Thien Hue province in Vietnam (9.0%) [9]. These differences may be due to from the different consumption patterns of alcohol between the population living in these provinces, concerns with the accuracy of this information based on self-reported data, or due to differences in the sociodemographic characteristics of the study populations.

**Physical inactivity.** In terms of physical activity, study participants spent about 28 MET-hours per week on physical activity, which is similar to results from the STEPs survey in

Vietnam in 2015 [24]. On the other hand, nearly one-quarter of study participants failed to achieve the WHO weekly recommendation of spending more than 600 MET-minutes in regular physical activities. The prevalence of physically inactive behavior in our study was markedly lower than the results from a previous study among 840 freshman students from the Vietnam National University in 2017 (44.0%) [21]. These differences may be explained by differences in the sociodemographic and other characteristics of the respective study populations: the previous study included 840 freshmen from the Vietnam National University, while our study population included individuals who were either farmers or people who worked in different sectors which may require more physical activity.

**Fruit and vegetable intake.** The number of fruits and vegetables consumed on a daily basis among the hypertensive adults in the present study was lower than has been recommended and more than three-quarters of the study population did not meet the most recent dietary recommendations of the WHO. The percentage of hypertensive patients with insufficient fruit and vegetable consumption is similar to the results from previous study in Vietnam [21, 25]. A study conducted on 369 adults aged 18 years and older in Quang Xuong, Thanh Hoa province, Vietnam in 2019 found that 72% of study participants who were diagnosed with HTN consumed insufficient amounts of fruit and vegetable [25]. The lack of counsel of the beneficial effects on eating fruits and vegetables could be the cause of the low fruit and vegetable consumption on a daily basis among hypertensive patients. Interventions that encourage fruit and vegetable intake should be emphasized for patients with chronic diseases as they contain vitamins, minerals, and beneficial nutrients that can help slow disease progression.

**Factors associated with multiple unhealthy lifestyle practices.** We found that more than one-quarter of study participants had any two or more unhealthy lifestyle practices, suggesting that the prevalence of unhealthy behaviors is high among hypertensive patients living in this rural Vietnam province. The study results show that men were more likely to have any two or more unhealthy lifestyle practices compared with women. This could be explained by the fact that in Vietnam, the men were more likely to have single unhealthy lifestyle practices such as smoking and consuming excessive amounts of alcohol [21, 24, 26]. In addition, we found that adults who performed unpaid work or were unemployed were more likely to have multiple unhealthy lifestyle practices compared to those who were employed. Several previous studies have reported a similar relationship between unemployment and having a high prevalence of unhealthy lifestyle practices [27, 28]. The present study results also suggests that individuals with a higher level of education were less likely to have two or more unhealthy behaviors compared to individuals with lower education levels. Adults with more education may be more knowledgeable about health risk factors so that they can avoid various unhealthy behaviors, which was reported in previous studies [29–31]. More and creative educational strategies promoting healthy lifestyle practices are needed for the general population and for patients with chronic diseases.

## Study strengths and limitations

The main strength of our study was the use of recent data which were collected among a relatively large sample of patients with uncontrolled hypertension in a large rural province of Vietnam, using a standardized protocol and validated questionnaire administered by trained healthcare workers.

There are several potential limitations, however, that must be kept in mind in interpreting our study results. The information collected was based on participant's self-report via in-person interview. Inasmuch, potential misclassification and interview biases may have occurred. The study used baseline data from participants enrolled in a clinical trial with a set of inclusion

criteria, therefore, the generalizability of our findings may be limited to patients with similar characteristics in rural settings in Vietnam.

## Conclusions

We conclude that unhealthy lifestyle practices are common among patients with uncontrolled hypertension in Hung Yen province, Vietnam. Male sex, unpaid work or unemployed status, and low education level were associated with a greater number of unhealthy lifestyle practices in these patients. Newer effective interventions and educational programs encouraging patients with hypertension to modify their lifestyle practices are needed among adults living in rural provinces of in Vietnam.

## Supporting information

**S1 Fig. ROC curve of the final logistic regression model.**
(TIF)

## Author Contributions

**Conceptualization:** Phuong H. Nguyen.

**Formal analysis:** Phuong H. Nguyen.

**Funding acquisition:** Jeroan J. Allison, Oanh M. Tran, Duc A. Ha.

**Methodology:** Phuong H. Nguyen, Bo Wang.

**Supervision:** Robert J. Goldberg, Jeroan J. Allison, Diep B. Nguyen, Ha T. Tran, Oanh M. Tran, Duc A. Ha, Hieu L. Nguyen, Brittany A. Tran, Hoa L. Nguyen.

**Visualization:** Phuong H. Nguyen.

**Writing – original draft:** Phuong H. Nguyen.

**Writing – review & editing:** Robert J. Goldberg, Hoa L. Nguyen.

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
