## [Decision Letter · Decision Letter 0]

29 Jan 2024

PONE-D-23-35910Lifestyle practices and associated factors among adults with hypertension in Vietnam: Conquering Hypertension in Vietnam-solutions at the grassroots level studyPLOS ONE

Dear Dr. Nguyen,

Thank you for submitting your manuscript to PLOS ONE. After careful consideration, we feel that it has merit but does not fully meet PLOS ONE’s publication criteria as it currently stands. Therefore, we invite you to submit a revised version of the manuscript that addresses the points raised during the review process and you will find below in the comments to the authors section. 

We look forward to receiving your revised manuscript.

Kind regards,

Christophe Seroyer

Academic Editor

PLOS ONE

 [The parent trial of this article was supported by the US National Heart, Lung, and Blood Institute of the National Institutes of Health under award number 1U01HL138631-01. The authors who were funded included Dr. Jeroan J. Allison, Dr. Duc A. Ha, and Dr. Oanh M. Tran. The present study was supported by the “Training Program for Strengthening Capacity in Non-communicable Diseases- D43TW011394-01” which was funded by the National Institutes of Health/Fogarty International Center. The authors who were funded included Dr. Jeroan J. Allison and Dr. Hoa L. Nguyen.].  

[The parent trial of this article was supported by the US National Heart, Lung, and Blood Institute of the National Institutes of Health under award number 1U01HL138631-01. The authors who were funded included Dr. Jeroan J. Allison, Dr. Duc A. Ha, and Dr. Oanh M. Tran. The present study was supported by the “Training Program for Strengthening Capacity in Non-communicable Diseases- D43TW011394-01” which was funded by the National Institutes of Health/ Fogarty International Center. The authors who were funded included Dr. Jeroan J. Allison and Dr. Hoa L. Nguyen.]

  [The parent trial of this article was supported by the US National Heart, Lung, and Blood Institute of the National Institutes of Health under award number 1U01HL138631-01. The authors who were funded included Dr. Jeroan J. Allison, Dr. Duc A. Ha, and Dr. Oanh M. Tran. The present study was supported by the “Training Program for Strengthening Capacity in Non-communicable Diseases- D43TW011394-01” which was funded by the National Institutes of Health/Fogarty International Center. The authors who were funded included Dr. Jeroan J. Allison and Dr. Hoa L. Nguyen.].

**Comments to the Author**

1. Is the manuscript technically sound, and do the data support the conclusions?

Reviewer #1: No

Reviewer #2: Yes

2. Has the statistical analysis been performed appropriately and rigorously? 

Reviewer #1: No

Reviewer #2: Yes

3. Have the authors made all data underlying the findings in their manuscript fully available?

Reviewer #1: No

Reviewer #2: Yes

4. Is the manuscript presented in an intelligible fashion and written in standard English?

Reviewer #1: Yes

Reviewer #2: Yes

5. Review Comments to the Author

Reviewer #1: PONE-D-23-35910: statistical review

SUMMARY. This is a cross-sectional study that examines the prevalence of unhealthy lifestyle practices and associated factors among Vietnamese adults with uncontrolled hypertension. The data are extracted from the baseline of a cluster randomized trial study and the principal outcome is a binary index, obtained by dichotomizing the sum of unhealthy practices and regressed over a battery of covariates. Due to a series of shortcomings, I’m afraid that the statistical analysis must be re-done (see the major issue below). I’m also appending a couple of specific issues about missing values and model selection that should be addressed by the authors.

MAJOR ISSUE

1. The principal outcome, unhealthy lifestyle practices, is summarized by a binary indicator (�1 and �2 unhealthy practices). This approach is generally not recommended for several reasons. First, it assumes exchangeability between unhealthy practices: being a smoker and a heavy drinker is considered the same as begin physically inactive and with an unhealthy diet. Second, in the case of incomplete profiles (missing values; see specific issue no 1), there are not obvious ways to compute the indicator. Third, it is based on the arbitrary threshold 2 (why not 1 or 3?). A simple way to overcome these difficulties is a logistic regression model where the four lifestyle practices are conditionally independent given subject- specific random effects and regression coefficients are outcome-specific. This kind of mixed effects model is available in all major statistical software platforms and would better support the discussion in the section “Prevalence of unhealthy lifestyle practices”.

SPECIFIC ISSUES

1. Missing values. Nothing is said about the treatment of incomplete profiles. How were missing values treated?

2. Model selection. The selection of the variables to be included in the model rely on the p-value threshold 0.2 in a univariate analysis (as said in line 170). This is not a recommended approach, because the p-value of a variable changes when further variables are included in the model, due to the correlation between covariates. More robust approaches rely on stepwise selection methods that rely on comparing the AIC values obtained before and after discarding a variable.

Reviewer #2: The manuscript entiled "Lifestyle practices and associated factors among adults with hypertension in

Vietnam: Conquering Hypertension in Vietnam-solutions at the grassroots level study" done by Phuong H. Nguyen et al. is interesting in the field of hypertention related to lifestyle. However, there are some major issues which should be clarified by the authors.

1- The number of included study subjects is small and there is only one province was selected for studying. Thus, the resuts could not be considered as representative for whole country. The authors should change the title and the conclusion statment;

2- Other included study subjects' characteristics should be done in Table 1: Socio-economic status, Co-morbidities, Daily salt intake, Stress, and Sleep disturbances;

3- Other causes of hypertension such as stress or obstructive sleep apnea should be excluded and discussed;

4- The P value for each OR should be done;

5- The Figures of OR and ROC curve should be done to well illustrate the results;

6- Sleep hygene should be discussed as potential risk of hypertension and lifestyle modifiable factor

Minor issues:

1-Lines 147-149: Socio-demographic, medical history, and clinical factors that were potentially

associated with the primary study outcome included age group (25-54, 55-64, 67-

149 74, and >75 years) : lack of 65-66 years, please correct it;

2- Data should be presented as mean +/- SD or SE.

6. PLOS authors have the option to publish the peer review history of their article (what does this mean?). If published, this will include your full peer review and any attached files.

Reviewer #1: No

Reviewer #2: **Yes: **Sy Duong-Quy

---

## [Author Response · Author response to Decision Letter 0]

13 Mar 2024

PONE-D-23-35910

Lifestyle practices and associated factors among adults with hypertension in Vietnam: Conquering Hypertension in Vietnam-solutions at the grassroots level study

We thank both the Editors and reviewers for their comments. Below, we have attempted to address each of the reviewers’ comments in a point-by-point manner.

Editor comments:

1. Stating the role of the funders.

Response: The funders had no role in our study design, data collection and analysis activities, decision to publish, or preparation of the manuscript. We have addressed this issue in the cover letter.

2. Funding information: Remove funding-related text from the manuscript. Update funding statement.

Response: We have removed this from the body of the manuscript and have updated the funding statement in our cover letter.

3. Data availability statement. Confirm at this time whether or not submission contains all raw data required to replicate the results of your study. 

“De-identified participant data that were collected for this study can be obtained by contacting the corresponding author. Data, with an accompanying dictionary, will be made available, beginning with publication, after approval of a short proposal summarizing the analyses to be done.” 

We have added this sentence at the end of the revised manuscript. (Line 338)

Reviewer #1: 

SUMMARY: This is a cross-sectional study that examines the prevalence of unhealthy lifestyle practices and associated factors among Vietnamese adults with uncontrolled hypertension. The data are extracted from the baseline of a cluster randomized trial study and the principal outcome is a binary index, obtained by dichotomizing the sum of unhealthy practices and regressed over a battery of covariates. Due to series of shortcomings, I’m afraid that the statistical analysis must be re-done (see the major issue below). I’m also appending a couple of specific issues about missing values and model selection that should be addressed by the authors. 

MAJOR ISSUE

1. The principal outcome, unhealthy lifestyle practices, is summarized by a binary indicator (≤1 and ≥2 unhealthy practices). This approach is generally not recommended for several reasons. First, it assumes exchangeability between unhealthy practices: being a smoker and a heavy drinker is considered the same as begin physically inactive and with an unhealthy diet. Second, in the case of incomplete profiles (missing values; see specific issue no 1), there are not obvious ways to compute indicator. Third, it is based on the arbitrary threshold 2 (why not 1 or 3?). A simple way to overcome these difficulties is a logistic regression model where the four lifestyle practices are conditionally independent given subject-specific random effects and regression coefficients are outcome-specific. This kind of mixed effects model is available in all major statistical software platforms and would better support the discussion in the section “Prevalence of unhealthy lifestyle practices”. 

Response: We thank the reviewer for their very helpful comment. Although the unhealthy lifestyle practices are not exchangeable, the total number of unhealthy practices remains important to understand. While there is an extensive literature examining factors associated with each unhealthy practice described in our paper, there is a lack of studies examining the number of unhealthy lifestyle practices. We chose 2 as the cut-off point for this study based on the distribution of this variable (Table 1) 

Number of unhealthy lifestyle practices n %

None 120 17.9

One 368 54.8

Two 161 24.0

Three 20 3.0

Four 2 0.3

 We have also performed a multinominal logistic regression model for three category outcomes (0, 1, and >= 2 unhealthy lifestyle practices) and the results were consistent with the results from the binominal logistic regression for the binary outcome (<=1 vs >=2) (Tables below). However, since the number of study participants with no unhealthy lifestyle practices was relatively small (n=120) in the present study, and frequency for some categories (eg., Educational level- College/University, Hypertension level III) were low, some relative risk ratios (RRRs) yielded from the multinominal logistic regression model were not stable (with wide range). We, therefore, selected the binominal logistic regression model as the main analytical approach for this study. We have added the sentence below to the revised methods section.

 “We also performed a multinominal regression model for the three-category outcome (having 0, 1, and � 2 unhealthy lifestyle practices). The results from this modelling approach were consistent with those from the binominal logistic regression for the binary outcome ( � 1 vs �2). However, due to the relatively small sample size in the “0 unhealthy” group and low frequencies in some response categories, we combined participants with 0 and 1 unhealthy lifestyle. We reported results from the binominal logistic model.”. (Line 178)

 We will, however, consider including these Tables as appendices if the Editors think that they will be useful for journal readers. 

Table 2: Study participant’s characteristics according to the number of unhealthy lifestyle practices

 Number of unhealthy 

lifestyle practices p-value

 0 1 �2 

Age (mean�SD) (years) 66.2�8.9 66.3�8.7 68.1�9.5 0.061

Age group (years) (n, %) 

25-54 11 (9.2) 30 (8.1) 9 (4.9) 0.160

55-64 42 (35.0) 115 (31.3) 58 (31.7) 

65-74 43 (35.8) 162 (44.0) 70 (38.3) 

75 24 (20.0) 61 (16.6) 46 (25.1) 

Male (n, %) 40 (33.3) 162 (44.0) 100 (54.6) 0.001

Marital status (n, %) 

With a partner 93 (77.5) 294 (79.9) 143 (78.1) 0.81

Education level (n, %) 

Primary school or less 24 (20.0) 90 (24.5) 58 (31.9) 0.110

Secondary school 66 (55.0) 201 (54.8) 87 (47.8) 

High school 18 (15.0) 46 (12.5) 29 (15.9) 

College/University 12 (10.0) 30 (8.2) 8 (4.4) 

Occupation (n, %) 

Employed 59 (49.2) 183 (49.7) 65 (35.5) 0.002

Unpaid work/Unemployed 36 (30.0) 104 (28.3) 83 (45.4) 

Retired 25 (20.8) 81 (22.0) 35 (19.1) 

BMI category (n, %) 

Underweight 5 (4.2) 18 (4.9) 8 (4.4) 0.59

Normal 48 (40.0) 173 (47.0) 95 (51.9) 

Overweight 38 (31.7) 99 (26.9) 43 (23.5) 

Obese 29 (24.2) 78 (21.2) 37 (20.2) 

HTN level (n, %) 

Level I 56 (46.7) 177 (48.1) 76 (41.5) 0.012

Level II 45 (37.5) 156 (42.4) 70 (38.3) 

Level III 19 (15.8) 35 (9.5) 37 (20.2) 

 

Table 3: Factors associated with the number of unhealthy lifestyle practices: Results from a multinomial logistic regression model.

 RRR (95% CI) *

 1 �2

Age group (years) (n, %) 

25-54 1.0 1.0

55-64 1.09 (0.49-2.41) 1.63 (0.60-4.44)

65-74 1.35 (0.59-3.09) 1.32 (0.47-3.71)

75 0.81 (0.31-2.14) 1.05 (0.33-3.37)

Sex 

Female 1.0 1.0

Male 1.93 (1.20-3.09) 3.72 (2.17-6.39)

Education level 

Primary school or less 1.0 1.0

Secondary school 0.56 (0.30-1.05) 0.36 (0.18-0.72)

High school 0.46 (0.20-1.02) 0.41 (0.17-0.99)

College/University 0.43 (0.17-1.09) 0.18 (0.06-0.57)

Occupation 

Employed 1.0 1.0

Unpaid work/

Unemployed 0.90 (0.52-1.54) 2.11 (1.15-3.87)

Retired 0.99 (0.53-1.86) 1.21 (0.58-2.52)

Hypertension level 

Level I 1.0 1.0

Level II 1.11 (0.70-1.74) 1.15 (0.68-1.95)

Level III 0.50 (0.26-097) 1.13 (0.57-2.23)

The reference category in the multinomial logistic regression model is 0 unhealthy lifestyle practices.

*RRR = relative risk ratio

SPECIFIC ISSUES

1. Missing values. Nothing is said about the treatment of incomplete profiles. How were missing values treated?

Response: There were only 2 study participants who had missing values on the education variable because they refused to answer the question about their level of education achieved; these individuals were not included in the multivariable regression model. We added this information in the footnote to Table 2 (line 227).

2. Model selection. The selection of the variables to be included in the model rely on the p-value threshold 0.2 in a univariate analysis (as said in line 170). This is not a recommended approach, because the p-value of a variable changes when further variables are included in the model, due to the correlation between covariates. More robust approaches rely on stepwise selection methods that rely on comparing the AIC values obtained before and after discarding a variable.

Response: The focus of the paper was on examining factors associated with the outcome and we used univariate analysis to screen variables for possible inclusion in our multivariable-adjusted regression analyses. We chose a p value of 0.2 as the cut-off point for the univariate analysis to avoid excluding factors that may have been potentially associated with the principal study outcome of the number of unhealthy lifestyle practices. Stepwise selection methods are very useful when independent variables are highly correlated. Independent variables in our study are not highly correlated. 

Reviewer #2: 

Manuscript entitled “Lifestyle practices and associated factors among adults with hypertension in Vietnam: Conquering Hypertension in Vietnam-solutions at the grassroots level study” done by Phuong H. Nguyen et al. is interesting in the field of hypertension related to lifestyle. However, there are some major issues which should be clarified by the authors.

1. The number of included study subjects is small and there is only one province was selected for studying. Thus, the results could not be considered as representative for whole country. The authors should change the title and the conclusion statement.

Response: The “Conquering Hypertension in Vietnam-solutions at the grassroots level study” is the title of the parent study. We only mentioned this in our paper title to link our paper to the parent study. As requested, however, we have revised the conclusion statements in both the title, the abstract and the main body of the manuscript to reflect the reviewer’s concern. (Line 1, 50, 54, 332, and 336) 

2. Other included study subjects’ characteristics should be done in Table 1: Socio-economic status, Co-morbidities, Daily salt intake, Stress, and Sleep disturbances.

Response: Various sociodemographic characteristics and comorbidities are presented in Table 2 according to the number of unhealthy lifestyle practices. Data on stress or obstructive sleep apnea were not collected in the parent trial.

3. Other causes of hypertension such as stress or obstructive sleep apnea should be excluded and discussed.

Response: While we agree with the reviewer’s suggestion, data on stress or obstructive sleep apnea were not collected in the parent trial.

4. The P value for each OR should be done.

Response: We have added p-values in the revised manuscript (Table 3). (Line 240)

5. The Figures of OR and ROC curve should be done to well illustrate the results.

Response: We have added the ROC figure in the revised manuscript (Figure 1). (Line 244) 

6. Sleep hygiene should be discussed as potential risk of hypertension and lifestyle modifiable factors.

Response: While we agree with the reviewer’s suggestion, data on sleep hygiene were not collected in the parent trial.

Minor issues

1. Line 147-149: Socio-demographic, medical history, and clinical factors that were potentially associated with the primary study outcome included age group (25-54, 55-64, 67-74, and ≥75 years): lack of 65-66 years, please correct it.

Response: We have corrected this error in the revised manuscript. (Line 149)

2. Data should be presented as mean ± SD or SE.

Response: Several continuous variables, such as the amount of time spent on physical activity during a typical week and number of fruit and vegetable servings consumed in an average day, (Table 1) were summarized using medians and the IQR since their distributions were skewed.

---

## [Decision Letter · Decision Letter 1]

24 Apr 2024

Lifestyle practices and associated factors among adults with hypertension: Conquering Hypertension in Vietnam-solutions at the grassroots level study

PONE-D-23-35910R1

Dear Dr Nguyen,

We’re pleased to inform you that your manuscript has been judged scientifically suitable for publication and will be formally accepted for publication once it meets all outstanding technical requirements.

Kind regards,

Christophe Leroyer

Academic Editor

PLOS ONE

**Comments to the Author**

1. If the authors have adequately addressed your comments raised in a previous round of review and you feel that this manuscript is now acceptable for publication, you may indicate that here to bypass the “Comments to the Author” section, enter your conflict of interest statement in the “Confidential to Editor” section, and submit your "Accept" recommendation.

Reviewer #1: All comments have been addressed

2. Is the manuscript technically sound, and do the data support the conclusions?

Reviewer #1: (No Response)

3. Has the statistical analysis been performed appropriately and rigorously? 

Reviewer #1: (No Response)

4. Have the authors made all data underlying the findings in their manuscript fully available?

Reviewer #1: (No Response)

5. Is the manuscript presented in an intelligible fashion and written in standard English?

Reviewer #1: (No Response)

6. Review Comments to the Author

Reviewer #1: (No Response)

7. PLOS authors have the option to publish the peer review history of their article (what does this mean?). If published, this will include your full peer review and any attached files.

Reviewer #1: No

---

## [Editor Report · Acceptance letter]

28 May 2024

PONE-D-23-35910R1 

PLOS ONE

Dear Dr. Nguyen, 

I'm pleased to inform you that your manuscript has been deemed suitable for publication in PLOS ONE. Congratulations! Your manuscript is now being handed over to our production team.

Kind regards, 

on behalf of

Dr. Christophe Leroyer 

Academic Editor

PLOS ONE